# SARS-CoV-2 Infection in Cities from the Southern Region of Bahia State, Brazil: Analysis of Variables Associated in Both Individual and Community Level

**DOI:** 10.3390/v15071583

**Published:** 2023-07-20

**Authors:** Murillo Ferreira da Silva, Uener Ribeiro dos Santos, Fabrício Barbosa Ferreira, George Rego Albuquerque, Ana Paula Melo Mariano, Hllytchaikra Ferraz Fehlberg, Íris Terezinha Santos de Santana, Pérola Rodrigues dos Santos, Luciano Cardoso Santos, Laine Lopes Silva de Jesus, Karoline Almeida Piton, Beatriz Santos Costa, Beatriz Sena Moreira Gomes, Vinicius Moreira Porto, Emanuelly da Silva Oliveira, Cibele Luz Oliveira, Renato Fontana, Bianca Mendes Maciel, Mylene de Melo Silva, Lauro Juliano Marin, Sandra Rocha Gadelha

**Affiliations:** 1Laboratório de Farmacogenômica e Epidemiologia Molecular, Universidade Estadual de Santa Cruz, Ilhéus 45662-900, Brazil; murilo_ty@hotmail.com (M.F.d.S.); fabriciob.lisboa@hotmail.com (F.B.F.); gralbu@uesc.br (G.R.A.); ferrazhellen@hotmail.com (H.F.F.); iristerezinha@gmail.com (Í.T.S.d.S.); luciano.cardoso23@hotmail.com (L.C.S.); llsjesus.bio@uesc.br (L.L.S.d.J.); kapiton.bio@uesc.br (K.A.P.); biascosta.sc@gmail.com (B.S.C.); beatriz.senamg@gmail.com (B.S.M.G.); vinii82@hotmail.com (V.M.P.); esoliveira.bio@uesc.br (E.d.S.O.); cibeleoliveira09876@gmail.com (C.L.O.); rfontana@uesc.br (R.F.); bmmaciel@uesc.br (B.M.M.); mmsilva@uesc.br (M.d.M.S.); lajumarin@hotmail.com (L.J.M.); 2Pós-Graduação em Ciências da Saúde, Universidade Estadual de Santa Cruz, Ilhéus 45662-900, Brazil; 3Laboratório de Imunobiologia, Universidade Estadual de Santa Cruz, Ilhéus 45662-900, Brazil; uener_edu@yahoo.com.br; 4Programa de Pós-Graduação em Ciência Animal, Universidade Estadual de Santa Cruz, Ilhéus 45662-900, Brazil; 5Departamento de Ciências Biológicas, Universidade Estadual de Santa Cruz, Ilhéus 45662-900, Brazil; 6Departamento de Ciências da Saúde, Universidade Estadual de Santa Cruz, Ilhéus 45662-900, Brazil

**Keywords:** COVID-19, SARS-CoV-2, epidemiology, risk factor, community risk, socio-economic status

## Abstract

The COVID-19 pandemic, caused by the severe acute respiratory syndrome coronavirus 2 (SARS-CoV-2), challenged public health systems worldwide. Individuals in low-income countries/regions are still at individual and community risk concerning inequality, sanitation, and economic conditions. Besides, during the pandemic, the transmission in municipalities and communities in the countryside and less developed regions kept viral spread and required structured and strengthened clinical and laboratory surveillance. Here, we present an observational, analytic, cross-sectional study conducted using secondary data from the Laboratório de Farmacogenômica e Epidemiologia Molecular (LAFEM)-Universidade Estadual de Santa Cruz (UESC), to evaluate individual and community factors associated to SARS-CoV-2 infection in outpatients from different cities from Southern Region of Bahia State, in Brazil. The data were collected between June 2021 and May 2022. The SARS-CoV-2 positivity by RT-qPCR was correlated with low socio-economic indicators, including the Human development index (*HDI_c_*) and Average worker salary (*AWSc*). Besides, in general, females were less likely to test positive for SARS-CoV-2 (OR = 0.752; CI 95% 0.663–0.853; *p* < 0.0001), while brown individuals had more positivity for infection (*p* < 0.0001). In addition, those who had clinical symptoms were more likely to test positive for SARS-CoV-2 (OR = 6.000; CI 95% 4.932–7.299; *p* < 0.0001). Although dry cough, headache, and fever were the most frequent, loss of taste (OR = 5.574; CI 95% 4.334–7.186) and loss of smell (OR = 6.327; CI 95% 4.899–8.144) presented higher odds ratio to be positive to SARS-CoV-2 by RT-qPCR. Nonetheless, the distribution of these characteristics was not homogenous among the different cities, especially for age and gender. The dynamic of SARS-CoV-2 positivity differed between cities and the total population and reinforces the hypothesis that control strategies for prevention needed to be developed based on both individual and community risk levels to mitigate harm to individuals and the health system.

## 1. Introduction

The COVID-19 pandemic, caused by the severe acute respiratory syndrome coronavirus 2 (SARS-CoV-2), challenged the public health system worldwide [1,2]. Until May 2023, more than 764 million cases and 6.9 million deaths were reported worldwide. Since the first case was reported in February 2020, there have been 37.4 million confirmed cases in Brazil, including 701 thousand deaths reported. Only in Bahia State, 1.5 million cases with 36 thousand deaths have been confirmed [3,4]. Bahia (BA) is one of the 26 States in Brazil, located in the northeast region of the country (11°24′35.5464″ S, 41°16′51.0852″ W) and divided into seven mesoregions and represents the fourth most populous State. Most cases of COVID-19 in Bahia were concentrated in the capital city of the State, Salvador, and then spread into regions outside the capital, reaching all cities and regions of the State and turning the Southern Region into one of the most affected areas after the capital.

The tendency for the infection to move toward the remote regions of the countries has posed challenges for public health systems in developing countries [4]. To understand this tendency, investigators addressed to know the impact of individual and community factors on the risk of viral infection and related diseases [5,6,7,8]. Individual levels include known risk factors, such as comorbidities, age, and gender. In turn, community level, including factors such as lower education, higher household crowding, and occupancy related to census tract level, were also determinants for virus spread [5,8]. Although Area Deprivation Index had been associated with seropositivity for SARS-CoV-2, the individual-level variables can attenuate this seropositivity at the community level [5].

The large national Brazilian territory made it difficult the control the pandemic. In Bahia State, social aspects, such as social inequalities, people living without access to water, precarious conditions of housing, and sanitation contributed to high infection rates [9]. Campaign hospitals, laboratories, and even public universities were required to support testing and treating COVID-19 patients [9]. Additionally, vigilance strategies were used to contain community spread and, together with the vaccines, helped to decrease transmission, symptoms, clinical manifestations, and worse outcomes of the patients [9,10,11,12].

Despite most SARS-CoV-2 infections are asymptomatic, many individuals present mild-to-moderate (fever, cough, and fatigue) or severe infection [13]. The host-virus-immune system interaction is crucial for patient prognosis. Comorbidities and advanced age are risk factors that can contribute to the worst prognosis and more severe clinical manifestations [13,14,15]. Interestingly, the viral load can be similar between asymptomatic and symptomatic individuals [16]. Nonetheless, severe infections were observed in patients without comorbidities and even in young adults [13,14,15].

Another important point in the COVID-19 pandemic is the emerging variants. The new variants can increase the viral capacity to spread and may have an impact on vaccine performance, disease severity, and diagnostic tools [17,18,19,20,21,22]. Additionally, social aspects, such as inequality, sanitation, and economic conditions, favor that individuals in low-income countries/regions are still at individual and community risk of SARS-CoV-2 infection and worse outcomes [13]. Thus, the transmission in municipalities and communities in the countryside and less developed regions can keep viral spread and require increased and strengthened clinical and laboratory surveillance.

Here, we aimed to characterize the epidemiological and clinical profile of outpatients from the Southern Region of Bahia State, in Brazil, that tested positive for SARS-CoV-2, between June 2021 and May 2022. We also analyzed risk factors for infection at both individual and community levels.

## 2. Materials and Methods

### 2.1. Study Design and Ethical Considerations

An observational, analytic, cross-sectional study was conducted using secondary data from the Laboratório de Farmacogenômica e Epidemiologia Molecular (LAFEM) database, located at Universidade Estadual de Santa Cruz (UESC). LAFEM/UESC worked in partnership with the Central Public Health Laboratory Professor Gonçalo Moniz (LACEN–BA), supporting the routine diagnostics for SARS-CoV-2 detection in the Southern Region of Bahia State, which was one of the main epicenters of the COVID-19 pandemic after the capital Salvador and its metropolitan region. The study was approved by Research Ethics Committee (Comitê de Ética em Pesquisa—CEP)/UESC, under protocol number CAAE: 39142720.5.0000.5526.

### 2.2. Laboratory Diagnosis of SARS-CoV-2 Infection

Nasopharyngeal swab specimens were collected from all individuals and tested at LAFEM/UESC by RT-qPCR. The viral RNA was extracted using an automated Loccus EXTRACTA 32 device and MVXA-P016 kit. The Allplex^TM^ 2019-nCoV assay (Seegene^®^, Seoul, Republic of Korea) and SARS-CoV-2 EDx assay (Bio-Manguinhos/FIOCRUZ, Rio de Janeiro, Brazil) were used to detect SARS-CoV-2 RNA by RT-qPCR, according to manufacturer’s protocol, using 7500 Fast Real-Time PCR System (Applied Biosystems^TM^, Life Technologies, Carlsbad, CA, USA). The results were classified as SARS-CoV-2 detected (positive), not detected (negative), and inconclusive.

### 2.3. Data Curation

Since 2020, LAFEM/UESC works in partnership with the LACEN–BA to support the routine diagnosis for SARS-CoV-2 detection in the Southern Region of Bahia State. From June 2021 to May 2022, 10,632 individuals from Municipal Health Departments (MHD)—outpatients care—and Hospitals from different cities in the Southern Region of Bahia State were tested. In this study, we did not include data from hospitalized patients. We only included outpatients. In total, 5015 outpatients from 34 cities in the Southern Region of BA State were primarily included in the analysis (Appendix A).

To evaluate individual levels of chance to test positive for SARS-CoV-2 (age, sex, symptoms, self-declared color, comorbidities), we first screened the data to remove individuals with incomplete data, and inconclusive results for RT-qPCR detection. A total of 4978 individuals were subsequently included. To evaluate the community profile in each city, we applied the same criteria described above for the individual level and then submit the cities to a baseline of samples and excluded all cities with ≤30 samples.

We normalized the variables to perform the statistical analysis, as follows: (1) age of participants was normalized to year, and participants with <1 year old were grouped as zero (0); (2) the presence of comorbidities at the LAFEM/UESC database were collected as “yes”, “chronic” and “no”, and we classify all individuals with “chronic” as “yes”; (3) individuals with inconclusive results for SARS-CoV-2 detection were excluded; (4) to assess community profile of SARS-CoV-2 infection, after application of the baseline of samples (*n* > 30), we have included 12 cities in our analysis.

### 2.4. Statistical Analysis

#### 2.4.1. Epidemiological and Geographical Indicators

To assess the Demand index (*D_i_*) and the Positivity index (*P_i_*) of the exams by cities, we crossed the data from LAFEM/UESC and the data from Instituto Brasileiro de Geografia e Estatística (IBGE) census 2010 (www.ibge.gov.br, accessed on 16 June 2023) using the Equations (1) and (2), respectively. The IBGE is the Brazilian institute responsible for collecting, processing, and disseminating demographic and population data for the implementation of public policies.

Equation (1): Demand index (*D_i_*)
(1)$$D_{i}=\left(\frac{E_{c}}{P}\right)\times 100$$
where *D_i_* represents the number of exams by cities according to total population of the respective city and are express as percentage; *E_c_* represents the number of exams by city; and *P* indicates the total population of the city according to IBGE census (www.ibge.gov.br, accessed on 16 June 2023).

Equation (2): Positivity index (*P_i_*)
(2)$$P_{i}=\left(\frac{E_{p}}{E_{c}}\right)\times 100$$
where *P_i_* represents the Positivity index of the exams express as percentage; *E_p_* represents the number of positive results by city; and *E_c_* represent the number of exams by city.

The *D_i_* and *P_i_* were geographically distributed by city using the QGIS^®^ version 3.26.3 Buenos Aires (QGIS Project, www.qgis.org, accessed on 16 June 2023) for Windows, a free and open source geographical system. Briefly, we used the Brazil and Bahia State cities free access maps in the format of shapes files (.shp) available on the IBGE website (www.ibge.gov.br, accessed on 16 June 2023) to cross data from *D_i_* and *P_i_* and geographical location. The cities were localized in the maps and the map was colored using QGIS^®^ version 3.26.3 Buenos Aires. Data from the Human development index (*HDI_c_*), Average worker salary (*AWS_c_*), and Schooling Rate between 6–14 years old (*SR_6–14_*) from each city were obtained from the IBGE census 2010 (www.ibge.gov.br, accessed on 16 June 2023). The income value for one minimum Brazilian wage is approximately US$ 265. A Pearson correlation analysis was performed to evaluate the correlation among the social indicators and *D_i_* and *P_i_* using GraphPad Prism software version 9.0.0 (GraphPad Prism Software. San Diego, CA, USA) for Windows. The respective correlation matrix figure was created using GraphPad Prism. Values of *p* < 0.05 were considered for statistical significance.

#### 2.4.2. Population Age Pyramid Construction and Age Analysis

To assess the graphical representation of the city’s age-sex distribution the individuals were stratified in groups: <10 years old, between 10 and 19 years old, between 20 and 29 years old, between 30 and 39 years old, between 40 and 49 years old, between 50 and 59 years old, between 60 and 69 years old, between 70 and 79 years old, between 80 and 89 years old, and ≥90 years-old. The age pyramids and overlapping curves were created using GraphPad Prism software version 9.0.0 (GraphPad Prism Software. San Diego, CA, USA) for Windows.

To assess differences between ages from different cities we evaluated the normal distribution of data by using the Kolmogorov-Smirnov test, with the Dallal-Wilkinson-Lilliefor corrected *p*-value. The data were analyzed using the Mann-Whitney test using GraphPad Prism software version 9.0.0 (GraphPad Prism Software. San Diego, CA, USA) for Windows. Data are presented as median and interquartile (IQR) and values of *p* < 0.05 were considered for statistical significance.

#### 2.4.3. Assessment of Factors Associated with SARS-CoV-2 Infection

Pearson’s Chi-squared test and Fisher’s exact test were used to determine the association between the frequency of each categorical variable and the detection of SARS-CoV-2 by RT-qPCR using GraphPad Prism software version 9.0.0 (GraphPad Prism Software. San Diego, CA, USA) for Windows. Values of *p* < 0.05 were considered for statistical significance. Data are presented as absolute frequency (n), percentage (%), odds ratio (OR), and 95% confidence interval (95% CI).

## 3. Results

### 3.1. Geographical Distribution of SARS-CoV-2 Infection in Cities from the Southern Region of Bahia State, Brazil

From June 2021 to May 2022, a total of 5015 outpatients from the Southern Region of BA State, who attended MHD and underwent SARS-CoV-2 detection by RT-qPCR at LAFEM/UESC were primarily included. After screening the data to remove individuals with incomplete data, and inconclusive results in the RT-qPCR, a total of 4978 individuals were included: 1400 (28.12%) SARS-CoV-2 positives and 3578 (71.88%) SARS-CoV-2 negatives. We distributed the samples by origin (Figure 1A,B) between the 34 cities included in the study (Appendix A). The cities that send more samples were: Itajuípe (*n* = 1108), Buerarema (*n* = 651), Itapé (*n* = 616), Floresta Azul (*n* = 585), and Pau Brasil (*n* = 524). Next, we established the Demand (*D_i_)* and Positivity (*P_i_*) index of SARS-CoV-2 infection from each location. The cities with more samples also presented high *D_i_*, such as Itapé (*D_i_* = 5.60%; *n* = 616), Floresta Azul (*D_i_* = 5.49%; *n* = 585), Itajuípe (*D_i_* = 5.26%; *n* = 1108), and Pau Brasil (*D_i_* = 4.83; Figure 1C). Nonetheless, *D_i_* differed from *P_i_* between the cities. As shown in Figure 1D, *P_i_* was higher for Pau Brasil (*P_i_* = 48.7%; *n* = 255), Arataca (*P_i_* = 42.3%; *n* = 55) and Itajuípe (*P_i_* = 36.9%; *n* = 409), but less than expected for Buerarema (*P_i_* = 23.35%) and Itapé (*P_i_* = 17.9%).

We also crossed the values from *D_i_* and *P_i_* of SARS-CoV-2 infection with sociodemographic data to observe a community risk level. We included data from the Human development index (*HDI_c_*), Average worker salary (*AWS_c_*), and Schooling Rate between 6–14 years old (*SR_6–14_*) from each city (Table 1). Overall, most cities presented low Human development index (*HDI_c_*), especially in cities with higher Positivity index (*P_i_*), such as Arataca (*HDI_c_* = 0.559 and *P_i_* = 42.31), and Itajuípe (*HDI_c_* = 0.559 and *P_i_* = 36.91; Table 1). Floresta Azul presented a combination of high *P_i_* (29.57), low *HDI_c_* (0.557), low Average worker salary (*AWS_c_* = 1.60), and low Schooling Rate between 6–14 years old (*SR_6–14_* = 92.80; Table 1). The *D_i_* was negatively correlated with *HDI_c_* (r = −0.59; *p* = 0.0442) and *AWS_c_* (r = −0.69; *p* = 0.0137), while *P_i_* was negatively correlated with *HDI_c_* (r = −0.61; *p* = 0.0339; Figure 1E; Appendix A).

### 3.2. Epidemiological Characteristics of Study Population

SARS-CoV-2 positive individuals (median 39 years, interquartile [IQR] 26–52) were older than SARS-CoV-2 negative (median 35 years, IQR 23–49; *p* < 0.0001; Table 2). As expected, according to the sociogeographical characteristic of Bahia State, our population consisted of manly female (*n* = 3125; 62.78%) and brown (*n* = 2776; 73.40%) individuals. In addition, females were less likely to test positive for SARS-CoV-2 (OR = 0.752; CI 95% = 0.663–0.853; *p* < 0.0001), while brown individuals were more likely to test positive for SARS-CoV-2 (*p* < 0.0001). Comorbidities did not increase the chance to test positive for SARS-CoV-2 in our study (OR = 1.174; CI 95% = 0.956–1.438; *p* = 0.1206). However, those who had clinical symptoms were more likely to test positive for SARS-CoV-2 (OR = 6.000; CI 95% = 4.932–7.299; *p* < 0.0001) compared to those that did not present symptoms.

Figure 2 shows the population pyramid with the frequency of individuals tested for SARS-CoV-2 by RT-qPCR. SARS-CoV-2 positive individuals were concentrated in the age group of 30–39 years and were female (Figure 2A), while the negative population was younger (between 20 and 29 years) (Figure 2B). Nonetheless, we observed a higher frequency of males in the group of SARS-CoV-2-positive individuals (Figure 2A,B). From the total population, the individuals were more concentrated in these two groups, between 20–29 and 30–39 years (Figure 2C). Figure 2D shows the overlapping curves, with a slight increase in males and a slight decrease in females in the SARS-CoV-2 positive group.

### 3.3. Comorbidities and Clinical Symptoms of SARS-CoV-2 Infection

As shown in Table 2, the frequency of comorbidities was low in our study (9.74%). The age in both groups, from SARS-CoV-2 positive and negative, partially explains the low frequency of comorbidities observed. Among individuals who had comorbidities, only diabetes (OR = 1.811; CI 95% 1.278–2.553; *p* = 0.0007), and kidney disease (OR = 4.271; CI 95% 1.143–16.16; *p* = 0.0444) increased de odds to test positive for SARS-CoV-2 (Table 3).

The frequency of symptoms according to SARS-CoV-2 detection is depicted in Table 4. The most common symptoms were dry cough (*n* = 2252; 47.11%), headache (*n* = 1782; 37.33%), fever (1583; 33.20%), and pharyngalgia (*n* = 1577; 33.06%; Table 4). Among reported symptoms, only diarrhea and abdominal pain were not associated with SARS-CoV-2 positivity. Besides, those who presented loss of taste (*p* < 0.0001) and loss of smell (*p* < 0.0001) had a higher odds ratio: 5.574 (CI 95% = 4.334–7.186) and 6.327 (CI 95% = 4.899–8.144), respectively, and they were more likely to test positive for SARS-CoV-2.

### 3.4. SARS-CoV-2 Infection at the Community Level in Cities from the Southern Region of Bahia State

To assess differences in SARS-CoV-2 infections among the cities, we applied a baseline of samples and excluded all cities with >30 samples. The cities that pass this criterion are presented in Table 1. First, we assessed the importance of gender for SARS-CoV-2 infection in each city (Figure 3). Although female was less likely to test positive for SARS-CoV-2 in the total population (OR = 0.752; CI 95% = 0.663–0.853; *p* < 0.0001; Table 2), this data was different when analyzing the data separately. Only in Itajuípe (OR = 0.647; CI 95% 0.504–0.833; *p* = 0.0006) and Itapé (OR = 0.613; CI 95% 0.407–0.935; *p* = 0.0215) cities, females were less likely to test positive for SARS-CoV-2. In other cities, gender was not correlated with SARS-CoV-2 infection. Important to highlight that these two cities (Itapé and Itajuípe) represented 35% of all individuals tested and included in our study and is reasonable to think that these cities had an impact on the epidemiology of SARS-CoV-2 infection in our study. In addition, Itajuípe and Itapé also presented high *P_i_*, 39.91 and 17.86, respectively (see Table 1).

To verify other factors involved, we analyzed the age distribution by each city (Figure 4). Statistically, the difference was only observed for Arataca [SARS-CoV-2 (+) median 37, IQR 25–45; SARS-CoV-2 (−) median 28, IQR 17–45; *p* < 0.05] and Pau Brasil cities [SARS-CoV-2 (+) median 42, IQR 29–57; SARS-CoV-2 (−) median 36, IQR 22.5–50.5; *p* < 0.001] with higher age for SARS-CoV-2 positive individuals (Figure 4).

## 4. Discussion

Here, we evaluated individual and community levels factors that could be related to SARS-CoV-2 positivity in cities from the Southern Region of Bahia State in Brazil, between June 2021 and May 2022. The general detection rate of SARS-CoV-2 was 28.12% in outpatients from 34 cities in this region. However, the Positivity (*P_i_*) index of SARS-CoV-2 infection among these cities was extremely diverse, even for neighboring cities, varying from 1.74% to 48.66%. Besides, most cities presented a low Human development index (*HDI_c_*), especially in cities with a higher Positivity index (*P_i_*).

About individual level factors, we observed that individuals testing positive for SARS-CoV-2 had differences in gender, age, self-declared color/Race, and presence of symptoms, compared to those that tested negative. Racial and ethnic disparities have been related to increased infections, hospitalizations, and deaths [23]. High seropositivity was observed in South America based on socio-economic status and the informal economic activities in low-income groups [24]. In fact, individuals with socio-economic disparity were affected disproportionally during the SARS-CoV-2 pandemic [25,26,27]. On the other hand, the COVID-19 pandemic can increase poverty and food insecurity in several countries [28].

As for gender, other investigators have shown higher positivity, the worst prognosis, and death outcomes for male compared to female patients [29]. Epidemiologically, one hypothesis is related to the higher level of exposure among men and a greater preventive care behavior of women with health. Biologically, hormones can influence the viral infection in host cells and impair the immune response to viral infection. Strogen may be protective against SARS-CoV-2 infection, while testosterone was associated with disease severity and high levels of serum cytokines in men [30].

Beyond gender, individuals between 20–39 years presented a high prevalence of infection compared to other age groups in the Southern Region of Bahia State. A study conducted by Wenzhou in China also presented a younger profile of infected people [31]. In fact, individuals between 20–39 years were more likely to leave home to work during the pandemic, and they tended to be more exposed to risks. This profile, however, differs from that presented before vaccination, in a previous study conducted for our group, analyzing individuals from the same region, that it was composed by older people (between 50–80 years old) [10]. It is worth mentioning that in this previous study, we analyzed only hospitalized individuals. Older people are more likely to have worse outcomes and need to be hospitalized, especially before vaccination for COVID-19. The vaccine decreased susceptible individuals from the networks of infection and decreased the prevalence of SARS-CoV-2 infection in the general population, including individuals with comorbidities. Besides that, older people were among the priority groups for vaccination.

We observed a low frequency of comorbidities in our study (only 9.74%), especially diabetes Mellitus (DM), and cardiovascular disease. DM impacts the vascular system, promoting endothelial dysfunction, or increasing disease severity and mortality by the immune system response. In turn, cardiovascular disease damages the cardiac tissue promoting an unbalance between metabolic demand and low heart rate. Besides, individuals with comorbidities present a high risk of severe COVID-19 [32,33,34]. A limitation of our study is the absence of patient outcome data to evaluate the impact of comorbidities on mortality rates in our population, and mortality rates by city.

Concerning symptoms, although dry cough, headache, and fever, were the most frequent symptoms, loss of taste (OR = 5.574; CI 95% 4.334–7.186) and loss of smell (OR = 6.327; CI 95% 4.899–8.144) presented higher odds ratio to test as positive to SARS-CoV-2. It is worth noting that these changes in taste and smell, even more, specific to COVID-19, are also common in other viral infections, such as those caused by influenza and rhinovirus. The most prevalent symptoms in this study are common to other viral infections. Headache, for example, is a highly prevalent symptom of COVID-19, but it is also a symptom reported in several acute viral respiratory infections. In addition, some hypotheses try to explain the mechanisms related to the development of these symptoms, including innate immune response to viral infection and production of several key cytokines [35,36].

Besides, we observed a slightly different pattern of symptoms comparing these data with the previous study conducted by our group in the Southern Region of Bahia State in 2020 with hospitalized patients, in which fever, dyspnea, and dry cough were most frequent [10]. Overall, all symptoms were less frequent than observed previously [10]. However, the previous study analyzed only hospitalized patients, who tend to be more symptomatic. Moreover, the introduction of vaccines in 2021, in Brazil and Bahia, and the emergence of new variants changed the profile of the infection as well as clinical presentation and mortality risk during the pandemic [37]. In this context, a limitation of our study was the absence of vaccination data for each patient, to assess the correlation between the vaccine (dose and time) and the manifestation of clinical symptoms. In relation to variants, in the beginning, it was observed rapid spread of SARS-CoV-2 in Brazil, together with the circulation of several viral variants of concern (VOCs) and variants under monitoring (VUMs), indicating the co-circulation of multiple SARS-CoV-2 lineages, linked to multiple importations [38]. In 2021, Gamma (P1) and Omicron (B.1.1.529) circulated in Brazil [38,39] and these variants caused recurrent pics of infections in different geographical locations [22]. Omicron has established itself as the most prevalent variant from the end of 2021 and has been classified into five main lineages, BA.1, BA.2, BA.3, BA.4, BA.5, and some sublineages (BA.1.1, BA.2.12.1, BA.2.11, BA.2.75, BA.4.6). In addition, Ômicron has been associated with higher transmissibility and higher risk of re-infection than other variants [39].

In Brazil, social factors such as the Human development index (*HDI_c_*), Average worker salary (*AWS_c_*), and Schooling rate between 6–14 years old (*SR_6–14_*) are indicators of social conditions of the communities that help us to understand the risk to viral and other infections. Here we observe a negative correlation between the Demand index (*D_i_*) with *HDI_c_*, and *AWS_c_*, and between the Positivity index (*P_i_*) with *HDI_c_*. These data reinforce the role of community risk level to understand the epidemiology of SARS-CoV-2 infection, once we observed an increase in SARS-CoV-2 positivity related to low social quality.

Finally, our data show that risk factors related to SARS-CoV-2 infection vary between the cities in the Southern Region of Bahia State. As an example, for the total population, we show that females were less likely to test positive for SARS-CoV-2, but when we analyzed the data by city, only Itajuípe and Itapé presented male as a risk factor. In addition, SARS-CoV-2-positive individuals were older, but, analyzing the data by city, only Arataca and Pau Brasil presented a significant difference between the age of individuals that tested positive and negative for SARS-CoV-2 infection. We suggest that multi-city epidemiology can help guide policies to combat the COVID-19 pandemic and other viral infections, but strategies at the community level can also help the creation of more effective health public policies.

## 5. Conclusions

Identifying the most vulnerable individuals can help design more effective prevention and control strategies to prevent unfavorable outcomes as result of SARS-CoV-2 and other viral infections. Individuals with respiratory symptoms were more likely to test positive, and the most frequent symptoms were fever, dry cough, and headache. Besides, in general, females were less likely to test positive for SARS-CoV-2. However, the dynamic of SARS-CoV-2 positivity differed among cities and the total population, and the positivity index was correlated with low socio-economic indicators. The data reinforce the hypothesis that control strategies for prevention needed to be developed based on both individual and community risk levels to mitigate harm to individuals and the health system.

## Figures and Tables

**Figure 1 viruses-15-01583-f001:**
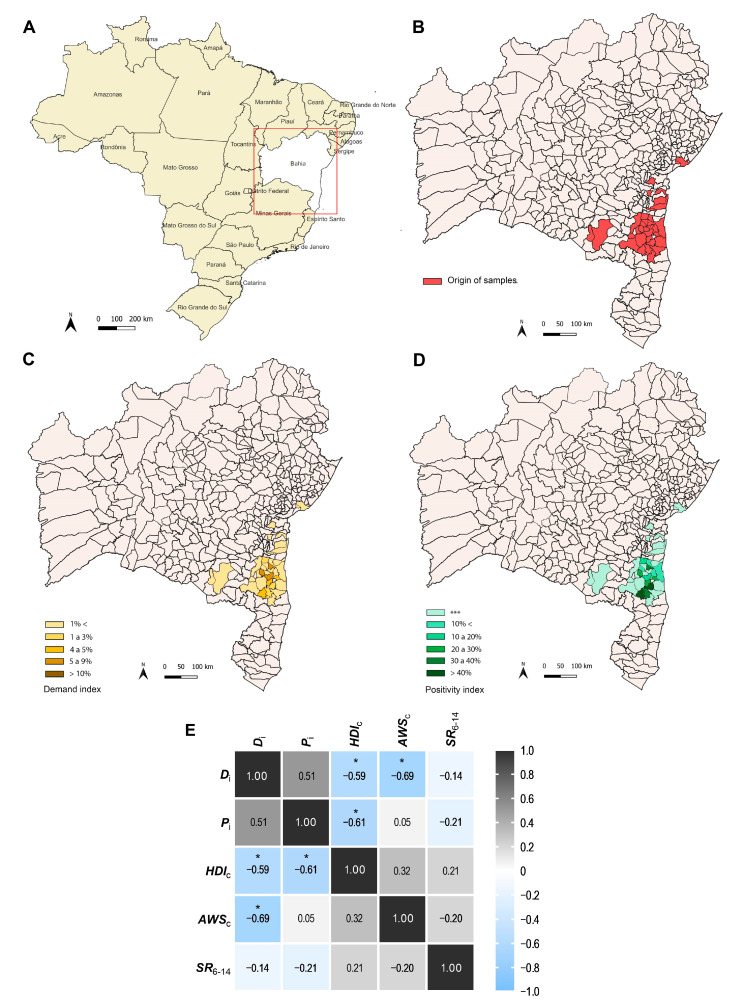
Geographical distribution of SARS-CoV-2 infections at the community level in cities from the Southern Region of Bahia State in Brazil. (**A**) The geographical location of Bahia State (highlighted in red) in Brazilian territory. (**B**) The territorial distribution of BA State; cities from Southern BA State included in the study are highlighted in red. (**C**) Demand index (*D_i_*—yellow gradient) and (**D**) Positivity index (*P_i_*—green gradient) of SARS-CoV-2 from each city included in the study are presented. (**E**) Correlation matrix of the community risk level. The value inside cells indicates the Person correlation. Asterix in the cells indicates significative *p* values with *p* < 0.05. *D_i_***,** Demand index; *P_i_*, Positivity index; *HDI_c_*, Human development index; *AWS_c_*, Average worker salary; *SR_6–14_*, Schooling rate between 6–14 years old; *** indicated the very low number of samples to calculate *P_i_*.

**Figure 2 viruses-15-01583-f002:**
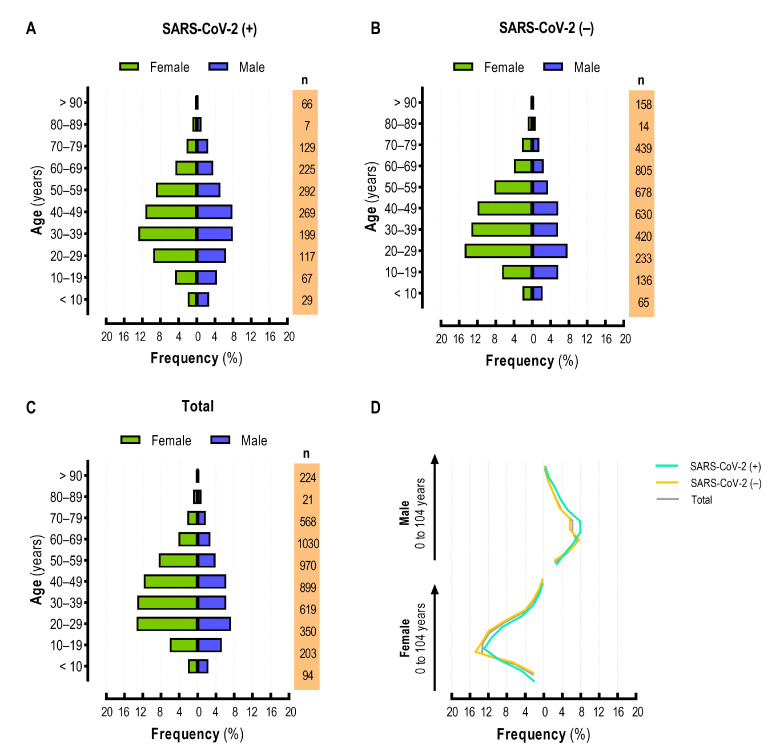
Population pyramid frequency from individuals tested for SARS-CoV-2 infection. (**A**) SARS-CoV-2 positive; (**B**) SARS-CoV-2 negative; (**C**) total population; (**D**) overlapping curves for population pyramid frequency.

**Figure 3 viruses-15-01583-f003:**
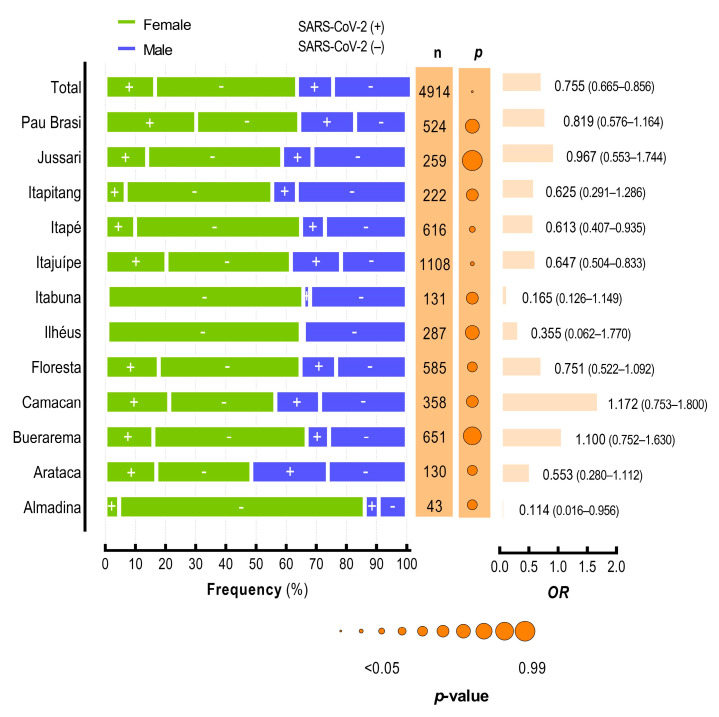
SARS-CoV-2 infection at the community level in cities from the Southern Region of Bahia State. Frequency of SARS-CoV-2 infection in Bahia cities according to gender. Female (green); male (blue); SARS-CoV-2 positive (+); SARS-CoV-2 negative (−); the total number (n) of individuals in each city is presented in the orange line; OR, odds ratio; values in parentheses indicate interval confidence 95% (CI 95%).

**Figure 4 viruses-15-01583-f004:**
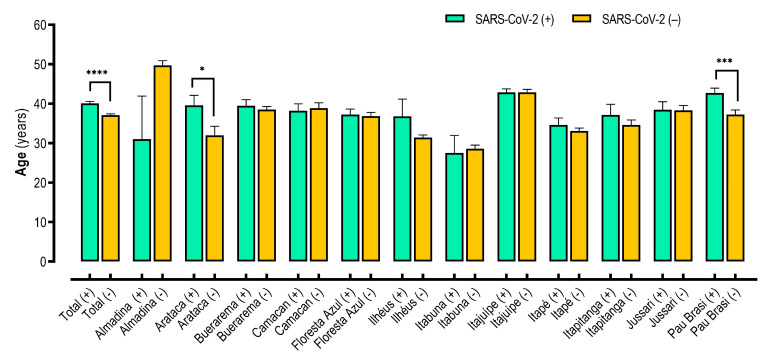
Age distribution by city in the Southern Region of Bahia State. Mann-Whitney test. Data are presented as mean ± standard error of the mean (S.E.M.). Vales of *p* < 0.05 were considered statistically significant. * *p* < 0.05; *** *p* < 0.001; **** *p* < 0.0001.

**Table 1 viruses-15-01583-t001:** The community factors associated with positivity for SARS-CoV-2 by RT-qPCR in cities from the Southern Region of Bahia State, Brazil.

City ^†^	*D_i_*	*P_i_*	*HDI_c_* ^‡^	*AWS_c_* ^‡^	*SR_6–14_* ^‡^
Almadina	0.68	9.30	0.563	2	95.30
Arataca	1.25	42.31	**0.559**	2.3	95.6
Buerarema	3.50	23.35	0.613	**1.70**	**91.50**
Camacan	1.14	36.03	0.581	2.00	**92.50**
Floresta Azul	5.49	29.57	**0.557**	**1.60**	**92.80**
Ilhéus	0.16	1.74	0.690	2.20	96.70
Itabuna	0.06	3.05	0.712	1.90	96.60
Itajuípe	5.26	36.91	**0.559**	1.80	96.30
Itapé	5.60	17.86	**0.559**	**1.60**	96.10
Itapitanga	2.17	14.86	0.571	**1.60**	97.00
Jussari	4.00	23.94	0.567	**1.70**	**94.50**
Pau Brasil	4.83	48.66	0.583	1.90	96.90

^†^ Cities with at least > 30 samples. ^‡^ Information recovered from Instituto Brasileiro de Geografia e Estatística (IBGE) census 2010 available at www.ibge.gov.br (accessed on 16 June 2023). Bold values indicate the smallest values. *D_i_*, Demand index. *P_i_*, Positivity index. *HDI_c_*, Human development index in the City. The *HDI_c_* ranges from 0 to 1. *AWS_c_*, Average worker salary in the city. The income value for one minimum Brazilian wage is approximately US$265. *SR_6–14_*, Schooling rate between 6–14 years old in the city. The *SR_6–14_* ranges from 0 to 100.

**Table 2 viruses-15-01583-t002:** Epidemiological Characteristics of the Study Population from the Southern Region of Bahia State, Brazil.

		SARS-CoV-2 N (%)			
Variables	Total *n* = 4978 (%)	SARS-CoV-2 (+) *n* = 1400 (%)	SARS-CoV-2 (−) *n* = 3578 (%)	OR (CI 95%)	*p*-Value
Age, median (IQR)	36 (23–50)	39 (26–52)	35 (23–49)	-	<0.0001 ^a^
Gender					
Female	3125 (62.78)	811 (16.29)	2314 (46.48)	0.752 (0.663–0.853)	<0.0001 ^b^
Male	1853 (37.22)	589 (11.83)	1264 (25.39)		
Self-declared color/Race					
Yellow	151 (3.99)	39 (1.03)	112 (2.96)	-	<0.0001 ^b^
White	382 (10.10)	106 (2.80)	276 (7.30)		
Indigenous	99 (2.62)	54 (1.43)	45 (1.19)		
Brown	2776 (73.40)	748 (19.78)	2028 (53.62)		
Black	374 (9.89)	73 (1.93)	301 (7.96)		
Not related *	1196	380	816		
Comorbidities					
Yes	485 (9.74)	151 (3.03)	334 (6.71)	1.174 (0.956–1.438)	0.1206 ^b^
No	4493 (90.26)	1249 (25.09)	3244 (65.17)		
Symptoms					
Yes	3553 (71.37)	1278 (25.67)	2275 (45.70)	6.000 (4.932–7.299)	<0.0001 ^b^
No	1425 (28.63)	122 (2.45)	1303 (26.18)		

^a^ Mann-Whitney test. ^b^ Pearson Chi-Squared test. * Not included in statistical analysis. Abbreviations: OR, odds ratio; CI 95%, 95% Confidence Interval; IQR, interquartile.

**Table 3 viruses-15-01583-t003:** Distribution of comorbidities in the study population from the Southern Region of Bahia State, Brazil.

		SARS-CoV-2 N (%)			
Variables	Total *n* = 4978 (%)	SARS-CoV-2 (+) *n* = 1400 (%)	SARS-CoV-2 (−) *n* = 3578 (%)	OR (CI 95%)	*p*-Value
Diabetes					
Yes	134 (2.69)	55 (1.10)	79 (1.59)	1.811 (1.278–2.553)	**0.0007** ^b^
No	4844 (97.31)	1345 (27.02)	3499 (70.29)		
Cardiovascular disease					
Yes	266 (5.39)	88 (1.71)	178 (3.58)	1.281 (0.983–1.663)	0.0645 ^b^
No	4712 (94.61)	1312 (26.36)	3400 (68.30)		
Immunodeficiency					
Yes	6 (0.12)	0 (0.0)	6 (0.12)	0.000 (0.000–1.787)	0.1942 ^c^
No	4972 (99.88)	1400 (28.12)	3572 (71.76)		
Kidney disease					
Yes	8 (0.16)	5 (0.10)	3 (0.06)	4.271 (1.143–16.16)	**0.0444** ^c^
No	4970 (99.84)	1395 (28.02)	3575 (71.82)		
Lung disease					
Yes	38 (0.76)	7 (0.14)	31 (0.62)	0.5750 (0.245–1.250)	0.1817 ^b^
No	4940 (99.24)	1393 (27.98)	3547 (71.25)		
HIV					
Yes	0 (0.0)	0 (0.0)	0 (0.0)	2.558 (infinity)	> 0.9999 ^c^
No	4978 (100.0)	1400 (28.1)	3578 (71.88)		
Cancer					
Yes	2 (0.04)	0 (0.0)	2 (0.04)	0.000 (0.000–5.256)	>0.9999 ^c^
No	4976 (99.96)	1400 (28.1)	3576 (71.84)		
Other					
Yes	130 (2.61)	31 (0.62)	99 (1.99)	0.7958 (0.524–1.202)	0.3227 ^b^
No	4848 (97.39)	1369 (27.50)	3479 (69.89)		

^b^ Pearson Chi-Squared test. ^c^ Fisher’s Exact test. Bold values indicate statistical significance. Abbreviations: OR, odds ratio; CI 95%, 95% Confidence Interval.

**Table 4 viruses-15-01583-t004:** Frequency of symptoms in the study population from the Southern Region of Bahia State, Brazil.

		SARS-CoV-2 N (%)			
Variables	Total *n* = 4978 (%)	SARS-CoV-2 (+) *n* = 1400 (%)	SARS-CoV-2 (−) *n* = 3578 (%)	OR (CI 95%)	*p* Value ^b^
Fever					
Yes	1583 (33.20)	721 (15.12)	862 (18.08)	3.256 (2.854–3.709)	**<0.0001**
No	3185 (66.80)	651 (13.65)	2534 (53.15)		
Not related *	210	28	182		
Fatigue					
Yes	212 (4.47)	107 (2.26)	105 (2.21)	2.670 (2.015–3.507)	**<0.0001**
No	4532 (95.53)	1252 (26.39)	3280 (69.14)		
Not related *	234	41	193		
Dry cough					
Yes	2252 (47.11)	860 (17.99)	1392 (29.12)	2.399 (2.110–2.727)	**<0.0001**
No	2529 (52.89)	518 (10.83)	2011 (42.06)		
Not related *	197	22	175		
Myalgia					
Yes	128 (2.70)	68 (1.44)	60 (1.26)	2.905 (2.032–4.113)	**<0.0001**
No	4621 (97.30)	1297 (27.31)	3324 (69.99)		
Not related *	229	35	194		
Dyspnea					
Yes	366 (7.72)	127 (2.68)	239 (5.04)	1.357 (1.084–1.699)	**0.0076**
No	4379 (92.28)	1232 (25.96)	3147 (66.32)		
Not related *	233	41	192		
Pharyngalgia					
Yes	1577 (33.06)	552 (11.57)	1025 (21.49)	1.556 (1.373–1.784)	**<0.0001**
No	3193 (6.94)	817 (17.13)	2376 (49.81)		
Not related *	208	31	177		
Diarrhea					
Yes	213 (4.47)	73 (1.53)	140 (2.94)	1.312 (0.984–1.745)	0.0659
No	4543 (95.53)	1292 (27.17)	3251 (68.36)		
Not related *	213	35	187		
Headache					
Yes	1782 (37.33)	663 (13.89)	1119 (23.44)	1.900 (1.673–2.158)	**<0.0001**
No	2991 (62.67)	711 (14.90)	2280 (47.77)		
Not related *	205	26	179		
Abdominal pain					
Yes	27 (0.57)	8 (0.17)	19 (0.40)	1.050 (0.485–2.325)	0.9084
No	4716 (99.43)	1350 (28.46)	3366 (70.97)		
Not related *	235	42	193		
Rhinorrhea					
Yes	1494 (31.33)	509 (10.67)	985 (20.65)	1.453 (1.282–1.658)	**<0.0001**
No	3275 (68.67)	859 (18.01)	2416 (50.66)		
Not related *	209	32	177		
Body ache					
Yes	230 (4.84)	107 (2.25)	123 (2.59)	2.264 (1.739–2.962)	**<0.0001**
No	4518 (95.16)	1254 (26.41)	3264 (68.74)		
Not related *	230	39	191		
Loss of taste					
Yes	297 (6.26)	197 (4.15)	100 (2.11)	5.574 (4.334–7.186)	**<0.0001**
No	4446 (93.74)	1161 (24.48)	3285 (69.26)		
Not related *	235	42	493		
Loss of smell					
Yes	302 (6.36)	208 (4.38)	94 (1.98)	6.327 (4.899–8.144)	**<0.0001**
No	4442 (93.63)	1151 (24.26)	3291 (69.37)		
Not related *	234	41	193		
Other					
Yes	440 (8.84)	145 (2.91)	295 (5.93)	1.286 (1.040–1.587)	**0.0182**
No	4538 (91.16)	1255 (25.21)	3283 (68.95)		

^b^ Pearson Chi-Squared test. * Not included in statistical analysis. Bold values indicate statistical significance. Abbreviations: OR, odds ratio; CI 95%, 95% Confidence Interval.

## Data Availability

Restrictions apply to the availability of these data. Data were obtained from “Observatório Lafem COVID Bahia” and are available at https://www.unbdatalab.org/uesclafemcovid (accessed on 16 June 2023), with the permission of DataLab—Universidade de Brasília—Brazil.

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
