# Peer review of "SARS-CoV-2 Infection in Cities from the Southern Region of Bahia State, Brazil: Analysis of Variables Associated in Both Individual and Community Level"

_viruses, 2023, doi:10.3390/v15071583_

Round 1

Reviewer 1 Report

In this study, Ferreira da Silva et al. aimed to analyze variables associated with SARS-CoV-2 infection at both the individual and community levels in cities from the Southern Region of Bahia State, Brazil.

The article is well-structured and well-designed.

Comments:

The manuscript could benefit from improvements in English language usage.

The resolution of Figure 1 should be enhanced.

It is recommended that the authors include all inputs, outputs, and scripts used to generate the figures in a publicly accessible repository, such as GitHub. This step is crucial for ensuring data reproducibility.

In this study, Ferreira da Silva et al. aimed to analyze variables associated with SARS-CoV-2 infection at both the individual and community levels in cities from the Southern Region of Bahia State, Brazil.

The article is well-structured and well-designed.

Comments:

The manuscript could benefit from improvements in English language usage.

The resolution of Figure 1 should be enhanced.

It is recommended that the authors include all inputs, outputs, and scripts used to generate the figures in a publicly accessible repository, such as GitHub. This step is crucial for ensuring data reproducibility.

Author Response

In this study, Ferreira da Silva et al. aimed to analyze variables associated with SARS-CoV-2 infection at both the individual and community levels in cities from the Southern Region of Bahia State, Brazil.

The article is well-structured and well-designed.

  1. The manuscript could benefit from improvements in English language usage.

Author’s response: We had made grammatical corrections and improvements in the English language.

  1. The resolution of Figure 1 should be enhanced.

Author’s response: We apologize for the figure quality. We uploaded a new Figure 1 with high-quality resolution.

  1. It is recommended that the authors include all inputs, outputs, and scripts used to generate the figures in a publicly accessible repository, such as GitHub. This step is crucial for ensuring data reproducibility.

Author’s response: We thank you for the comment, and we agree with the reviewer about the reproducibility of data in science. In our work, we did not use software with source code to have inputs, outputs, and scripts that generate the figures, as the R project for Statistical Computing and their packages. We just use the GraphPad Prism software version 9.0.0 (GraphPad Prism Software. San Diego, CA, EUA) for Windows with a friendly interface that automatically generates the graphs/figures, and the user just needs to change the design as color, and size, for example. To create the representative maps (Figure 1), we used the QGIS® version 3.26.3 Buenos Aires (QGIS Project, www.qgis.org) for Windows. The QGIS is a free and open source geographical system and did not use source code and scripts to generate figures. The software has a friendly interface, and the user needs to open the shapes files (.shp) that contain the illustration of the map, select the State or City and add color to it. The shapes files are free and available on the IBGE website (www.ibge.gov.br). Besides, we rewrite the Statistical Analysis section (on page 5) to improve this information for the readers of the manuscript as follow bellow

The Di and Pi were geographically distributed by city using the QGIS® version 3.26.3 Buenos Aires (QGIS Project, www.qgis.org) for Windows, a free and open source geographical system. Briefly, we used the Brazil and Bahia State cities free access maps in the format of shapes files (.shp) available on the IBGE website (www.ibge.gov.br) to cross data from Di and Pi and geographical location. The cities were localized in the maps and the map was colored using QGIS® version 3.26.3 Buenos Aires.

Reviewer 2 Report

The original research presented to me for review deals with the very clinically relevant issue of analysis of variables associated to both individual and community levels during SARS-CoV-2 infection. It is a work based on a very large clinical material from many medical centers from Brazil.

The work is written in good language, the methodology and assumptions of the work are not questionable. The results are presented very clearly and the conclusions are logical to the results obtained.

However, before accepting the paper for publication, I would suggest some additions that may carry the value of the work:

1. regarding the symptoms of loss of smell and taste, it should be emphasized in the discussion that this is not a symptom specific only to COVID-19, it is often observed in other infections based on: PMCID: PMC8988454

2. headaches are frequently observed during COVID-19 infection and the infection itself may have long-term effects on the course of diseases such as migraine, it is worth mentioning in the discussion with regard to the potential mechanism of their induction based on: PMID: 35758225

3. has the course of the severity of the observed symptoms depending on the passage or absence of vaccination against SARS-CoV-2 been studied?

Author Response

The original research presented to me for review deals with the very clinically relevant issue of analysis of variables associated to both individual and community levels during SARS-CoV-2 infection. It is a work based on a very large clinical material from many medical centers from Brazil.

The work is written in good language, the methodology and assumptions of the work are not questionable. The results are presented very clearly and the conclusions are logical to the results obtained.

However, before accepting the paper for publication, I would suggest some additions that may carry the value of the work:

  1. Regarding the symptoms of loss of smell and taste, it should be emphasized in the discussion that this is not a symptom specific only to COVID-19, it is often observed in other infections based on: PMCID: PMC8988454
  2. Headaches are frequently observed during COVID-19 infection and the infection itself may have long-term effects on the course of diseases such as migraine, it is worth mentioning in the discussion with regard to the potential mechanism of their induction based on: PMID: 35758225

Author’s response: We agree with the reviewer that these two symptoms are not specific for COVID-19. To better clarify this statement, we include a new paragraph in our manuscript (on Page 15) discussing questions 1 and 2 of the reviewer with references. Please, see the paragraph below.

“It is worth noting that these changes in taste and smell, even more specific to Covid-19, are also common in other viral infections, such as those caused by influenza and rhinovirus. In fact, the most prevalent symptoms in this study are common to other viral infections. Headache, for example, is a highly prevalent symptom of Covid-19, but it is also a symptom reported in several acute viral respiratory infections. In addition, some hypotheses try to explain the related mechanisms related to the development of these symptoms, including innate immune response to viral infection and production of several key cytokines [35,36].”

  1. Has the course of the severity of the observed symptoms depending on the passage or absence of vaccination against SARS-CoV-2 been studied?

Author’s response: The reviewer’s comment is quite interesting. In our work, the samples and forms with data that are available at LAFEM/UESC database were received from different municipalities, and, unfortunately, we do not have vaccination data available to answer this question. To avoid misinterpretation, we have included a sentence in our manuscript (on Page 16) regarding the limitations of the study concerning vaccination and the absence of this data. Please, see the sentence below.

“A limitation of our study is the absence of vaccination data for each patient, to assess the correlation between the vaccine status (dose and time) and the manifestation of clinical symptoms.”

We thank you once again for the thorough review carried out by the reviewers. We hope this revised manuscript has addressed all concerns and look forward to your positive response soon.